# Can empathy be taught? A cross-sectional survey assessing training to deliver the diagnosis of end stage renal disease

Alice Doreille[1], Eve Vilaine[2], Xavier Belenfant[3,4], Wided Tabbi[3], Ziad Massy[2,5], Emmanuelle Corruble[6], Odile Basse[7], Yosu Luque[1,8], Eric Rondeau[1,8], Dan Benhamou[9,10], Helene François[1,8,10]*

1 Department of Nephrology and Transplantation, Hôpital Tenon, AP-HP, Paris, France, 2 Department of Nephrology, CHU Ambroise Paré, AP-HP, Paris, France, 3 Department of Nephrology, CHI André Grégoire, Montreuil, France, 4 Réseau de Néphrologie d'Ile de France (Rénif), Paris, France, 5 Centre for Research in Epidemiology and Population Health (CESP), UMRS 1018, team 5, UVSQ, University Paris Saclay, Villejuif, France, 6 Department of Psychiatry, Hôpital Bicêtre, AP-HP, Kremlin Bicêtre, France, 7 Association France Rein Ile de France, Paris, France, 8 Sorbonne Université, UMR_S1155, Paris, France, 9 Department of Anesthesiology, Hôpital Bicêtre, AP-HP, Kremlin Bicêtre, France, 10 LabForSIMS Simulation Center, Paris Sud University, Kremlin Bicêtre, France

* helene.francois@aphp.fr

**Data Availability Statement:** All relevant data are within the paper and its Supporting information files.

## Abstract

### Background

Receiving the diagnosis of kidney failure has a major impact on patients. Yet, the way in which this diagnosis should be delivered is not formally taught within our medical curriculum. To fill this gap we set up a training course of kidney failure diagnosis delivery for nephrology trainees since 2016. This study assessed the effectiveness of this educational intervention.

### Methods

The primary outcome was change in the empathy score immediately after the training session and several months afterward, based on the Jefferson Scale of Physician Empathy (JSPE). Self-reported change in clinical practice was also evaluated. As control groups, we assessed empathy levels in untrained nephrology trainees (n = 26) and senior nephrologists (n = 71). Later on (>6 months) we evaluated participants' perception of changes in their clinical practice due to the training.

### Results

Six training sessions permitted to train 46 trainees. Most respondents (76%) considered the training to have a durable effect on their clinical practice. Average empathy scores were not significantly different in pre-trained trainees (average JSPE: 103.7 ± 11.4), untrained trainees (102.8 ± 16.4; P = 0.81) and senior nephrologists (107.2 ± 13.6; P = 0.15). Participants' empathy score significantly improved after the training session (112.8 ± 13.9; P = 0.003). This improvement was sustained several months afterwards (average JSPE 110.5 ± 10.8; P = 0.04).

**Funding:** HF received funding from the Agence Regionale de la Sante en Ile de France The sponsor played no role in the design of the study, data collection and analysis, decision to publish or preparation of the manuscript.

**Competing interests:** NO authors have competing interests.

## Conclusion

A single 4-hour training session can have long lasting impact on empathy and clinical practice of participants. Willingness to listen, empathy and kindness are thought to be innate and instinctive skills, but they can be acquired and should be taught.

## Introduction

Traditionally, medical interest has been focusing on diseases and organ systems rather than on physician-patient relationship. There has however been a shift in paradigm amongst the medical community, thus making care more patient-centred, giving more importance to physicians' communication skills [1–3]. This is particularly important in medical fields where breaking bad news is daily practice. In oncology, it is well documented that this bad news delivery has a major impact on the patients' emotions and care [4]. In nephrology, severe chronic kidney disease (CKD) requiring replacement therapy is indeed a severe condition with lower 5-year survival than all cancers combined [3,5]. Patients consider the delivery of kidney failure diagnosis as bad news, which often shatters their life and environment and has major consequences in accepting treatments. It is indeed experienced by some patients as a trauma and has a major impact on patients' care [6]. Reducing the impact of this bad news delivery is possible through patients education where a good communication between physicians and patients is mandatory, but also by improving teaching to nephrology trainees [7]. In most countries, teaching of communication skills is seldom in the foreground and the "Breaking of bad news" is rarely formally taught to nephrology trainees [8]. This requires an in-depth questioning of our teaching practices in nephrology but also a rethinking of skills that need to be taught to trainees to be good physicians. Patient-centred care requires a willingness to listen, empathy and kindness. These skills are thought to be innate and instinctive. As a result, they are barely taught or even discussed in our medical curriculum. It therefore seems paramount to include teaching of these skills in trainees' training program of and evaluate its effectiveness and impact. To fill this training gap and in order to improve CKD patients' care, our regional Health Authority financed a training program on kidney failure diagnosis delivery. This project was highly encouraged by CKD patients' associations. We report here a cross-sectional survey assessing modification of empathy and impact of this training program on clinical practice regarding physician-patient communication in nephrology.

## Methods

### Teaching methodology

Since 2016, a single 4-hour training session was offered annually through a mailing list invitation for all trainees studying nephrology in the Paris area (either trainees completing their whole training in Paris or trainees from abroad completing a one-year training period in Paris). The training session was not compulsory and trainees participated if they felt it could be useful to them. The nephrology teaching committee of the Paris area approved the session. The sessions took place at LabForSIMS, a place dedicated to teaching through the use of simulation tools in Paris Sud University. Two senior nephrologists, an anaesthetist specialised in teaching through simulation, a psychologist and/or a psychiatrist trained in teaching patient-physician relation attended each session.

**Testimony from a patient regarding his perception of the delivery of a kidney failure diagnosis.**   A 15-minute video with a patient's testimony was played. This patient is an active member of the France Rein Association, a French patients' association. He has experienced denial of kidney failure, leading him to start dialysis with a catheter in context of metabolic emergency. In this video, the patient explains and reflects on the reasons of his denial and dialysis rejection. His personal story is a typical example of how insufficiently a patient can be prepared to renal replacement therapy and the physician-patient's communication lacks.

**Role-playing using three successive storylines.**   Five senior nephrologists and a specialist in simulation-based teaching devised three typical clinical scenarios. Representatives of a patients' association (France Rein) reviewed the scenarios and gave their feedback. The first scenario was about announcing kidney failure with the necessity to think about renal replacement therapy to a patient with a good follow-up and medical trust. The second scenario looked at the case of a patient with CKD displaying denial and refusing treatment, while experiencing extreme anxiety due to an autosomal dominant polycystic kidney disease. The last scenario related to an emergency situation, where the patient was hospitalised in the context of an undiagnosed kidney failure and needed to receive emergency dialysis. For each scenario, 3 trainees volunteered to play the respective roles of the patient, the patient's relative and the physician. The clinical scenario was explained to each of them aside. They then went into a consultation room nearby, where they freely played out the consultation with the scenario in mind. The consultation was broadcasted live to all the other participants in a separate room next to the consultation room. A debriefing session took place after each role-play to discuss various aspects of the physician-patient interaction and participants were encouraged to have a reflective thinking and provide feedback about their performance. A global debriefing concluded the training session to reflect, provide feedbacks about the session and explore participants' emotions.

## Teaching evaluation

**Evaluation of satisfaction.**   All participants were given an anonymous survey at the end of the training session to assess their satisfaction as well as the potential impact of the training session on their medical practice (S1 Item in S4 File). On spring 2019, an additional survey was sent out to all participants by email to assess the long-term impact of the training session (S2 Item in S4 File).

**Evaluation of empathy.**   Self-administered questionnaires were used to assess participants' empathy, based on the Jefferson Scale of Physician Empathy (JSPE), a 20-item physician self-assessment tool, which evaluates empathy on a 7-point scale (1 = strongly disagree, 7 = strongly agree). The scale consists of three components: perspective taking, compassionate care and standing in the patient's shoes [9,10].

To assess pre- and post-training empathy scores, all participants were invited to fill in this questionnaire prior to the training and immediately after. In order to assess post-training empathy levels on the longer term, we also sent out an online questionnaire to all participants during the spring of 2019. Over the same period of time, we sent out the questionnaire to all nephrology trainees (n = 110) and senior nephrologists of the Paris area (n = 140), to serve as control groups. Long-term impact was thus evaluated at least 6 months after the training session and ranged from 6 months to 34 months.

**Statement of ethics.**   All participants gave written consent for the use of their questionnaire and JSPE scores.

**Statistical analysis.**   Quantitative variables are expressed as means and standard deviation (SD) and the qualitative values as percentages. Analysis of variance (ANOVA) was used to

analyse the differences among group means. Pearson correlation was used to determine the correlation of continuous variables. Analyses were performed using R Statistical Software (Foundation for Statistical Computing, Vienna, Austria). Minimal Data set are available as supporting information S3 File.

## Results

Amongst the 110 nephrology trainees to whom the session was offered, 46 were trained over six sessions, from May 2016 to October 2018. Nineteen (41%) of them were women. Most participants were first year trainees (n = 24, 52%), the rest (n = 22, 48%) was evenly shared between trainees from second to fourth year.

### Satisfaction

Among 39 (out of 46, 85%) participants who filled in the satisfaction and impact survey immediately after receiving the training, 97% rated the training as either essential or very useful to their clinical practice and found the role-play essential (n = 19, 47.5%) to very useful (n = 19, 47.5%). In addition, 95% of respondents found the length of the training session to be appropriate.

### Impact

Immediately after the training session, all but one participant said they considered the training session would change their medical practice (shown in Fig 1a). On spring 2019, 28 (out of 46, 61%) participants answered the impact survey online. 76% of respondents said that the training session had changed their medical practice in the following weeks (shown in Fig 1b) and 69% considered that the training session still impact their medical practice (shown in Fig 1c). All of the 4 participants who answered "No" to either of these 2 questions commented that they still had not had to deal with delivering a diagnosis of terminal kidney disease. What emerged from the comments was respondents taking more time to deliver the diagnosis and allowing more space for patients' questions and concerns. They also felt more confident with the Breaking of bad news delivery.

### Empathy score

All training session participants (n = 46, 100%) completed the Jefferson Scale of Physician Empathy (JSPE) before the training session. Forty-four (out of 46, 96%) participants completed the JSPE immediately after the training. During the spring of 2019, out of the 110 nephrology trainees from the Paris area contacted, 54 completed the JSPE online. Twenty-eight of them had received the training 6 to 34 months before. The other twenty-six trainees did not attend the training. During the same period, 140 senior nephrologists were contacted, 71 of them took part in the study.

Empathy scores were not significantly different (102.8 ± 16.4 *versus* 103.7 ± 11.4, p-value = 0.81) in untrained and pre-training nephrology trainees respectively (shown in Fig 2). Post training, participants' empathy score significantly improved (112.8 ± 13.9, p = 0.003) and was sustained several months afterwards (110.5 ± 10.8, p = 0.04) (shown in Fig 2). Average empathy score for senior nephrologists (106.6 ± 13.5) was in between the untrained/pre-training trainees and the post-training trainees, with no significant difference (p = 0.15 and p = 0.19 respectively) (shown in Fig 2).

In the participants' cohort, empathy score were not correlated with gender or residence year. In the senior nephrologists' cohort, out of the 71 respondents, 34 were women (48%).

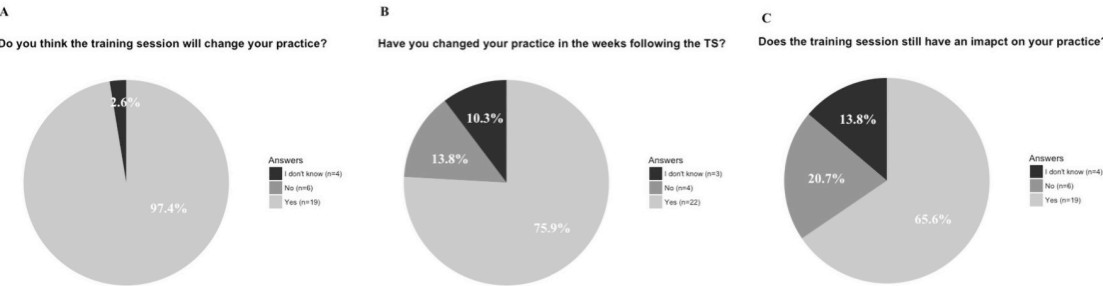

**Fig 1. Auto-evaluation of the impact of the training session on clinical practice. A**. Immediately after the training session the question "Do you think the training session will change your practice?" was anonymously asked. Thirty-eight respondents answered positively (97.4%), one participant answered "I don't know" (2.6%) and no participant answered negatively. **B. and C**. During spring 2019, 29 participants answered an online survey. Twenty-two (75.9%) of respondents have actually considered that the training session has changed their medical practices in the following weeks and twenty respondents (65.6%) think the training session still impact their medical practices.

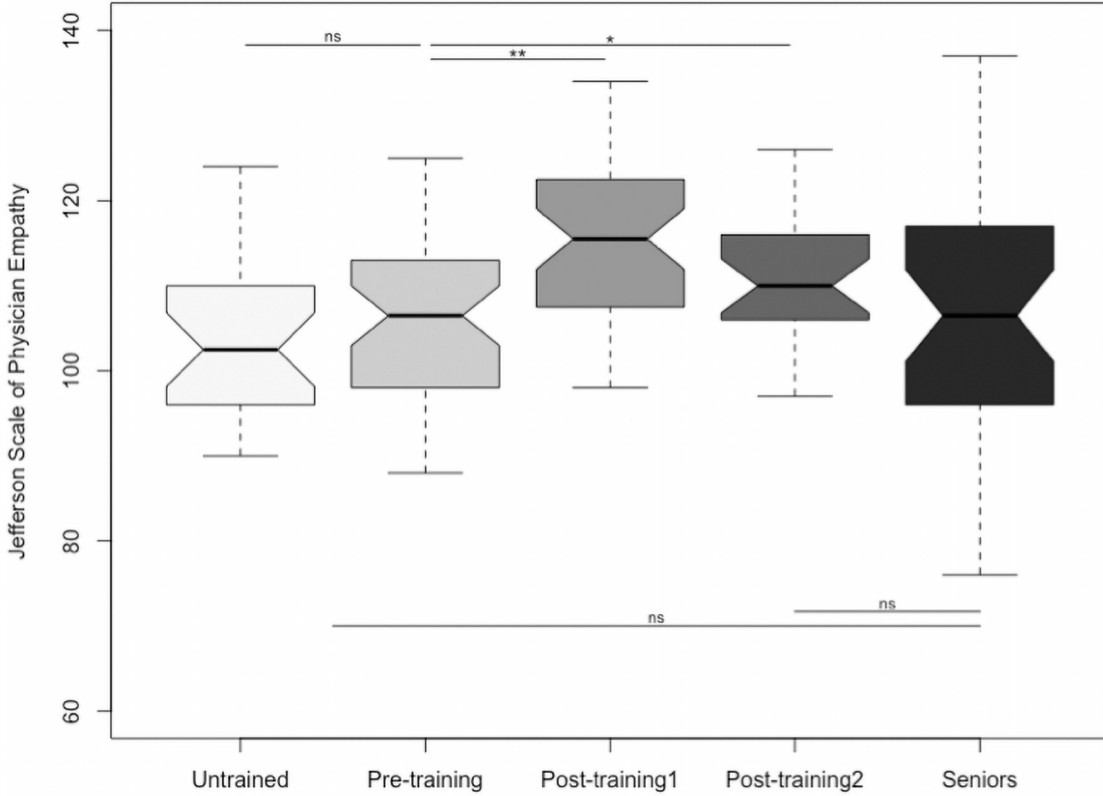

**Fig 2. Empathy in nephrology trainees (untrained, pre-training, post-training short and long term) and senior nephrologists.** Empathy scores using the JSPE were similar in untrained (with a mean of 102.8 and standard deviation of 16.4) and pre-training trainees (103.7 ± 11.4), p = 0.81. Post training, participants' empathy score significantly improved (112.8 ± 13.9) and was sustained several months afterwards (110.5 ± 10.8), respectively p = 0.003 and p = 0.04. Average empathy score for senior nephrologists (107.2 ± 13.6) was in between the untrained/pre-training trainees and the post-training trainees, with no significant difference (respectively p = 0.15 and p = 0.19). *Post-training1 corresponds to the evaluation made directly after the training session. Post-training2 corresponds to the evaluation made in spring 2019 (5 to 34 months after the training session). ns: Non significant difference,* $^*p<0.05$, $^{**}p<0.005$.

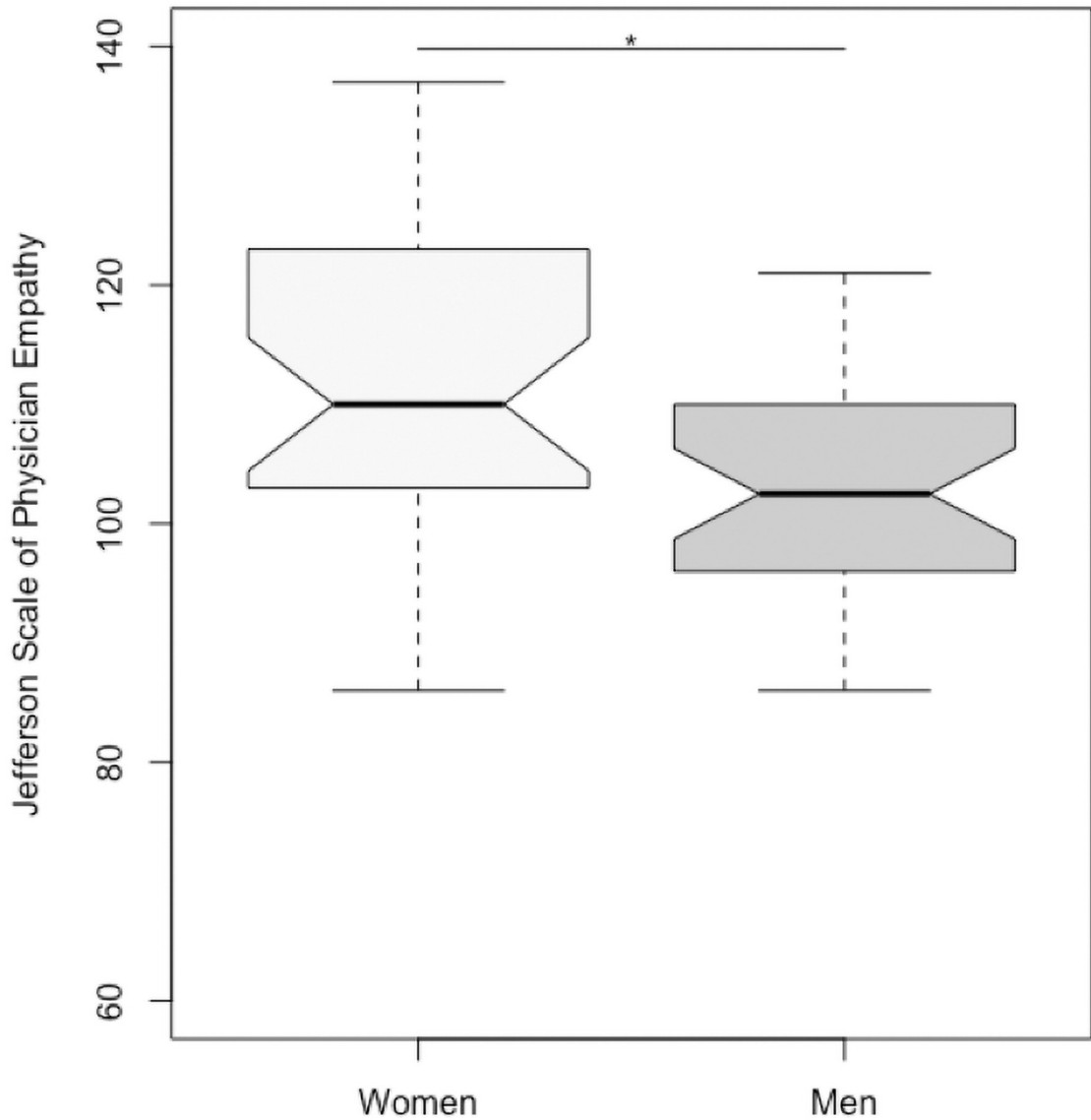

**Fig 3. Empathy in senior nephrologists according to gender.** Female nephrologists have significantly higher empathy score (111.1 ± 15.2) compare to male nephrologists (103.8 ± 11.2), p = 0.02. *$p<0.05$.

The mean age was 47 year old (SD 10.6). Most participants were either hospital practitioner (n = 38, 54%) or university and hospital physicians (n = 17, 24%). Six participants (8%) made practice in an associative structure and 10 (14%) in private structure. Female nephrologists had significantly higher empathy score (JSPE 111.1 ± 15.2 *versus* 103.8 ± 11.2, p = 0.02) reaching the post-training level of trainees (shown in Fig 3). No correlation was found between empathy score and age or modality of practice.

## Discussion

The main result of our study is that a single session had an immediate and long lasting impact on the medical practice of most participants. Furthermore, empathy in participants increased

significantly after the training session and the improvement was sustained several months afterward.

Several limitations to our study have to be raised. First, as the training was not compulsory, there may be significant discrepancies between the nephrology trainees who participated and those who did not. Yet there were no significant difference between pre-trained trainees' JSPE and untrained trainees' JSPE.

Our main results are based on self-assessed clinical impact and self-reported empathy for patients. There are undoubtedly weaknesses to such subjective evaluations. Yet the JSPE, which is quantitative self-reported evaluation of empathy, has been recognized as the most researched and widely used instrument in medical education research [10]. Numerous studies have validated JSPE as a robust empathy-measuring instrument [9–11]. Moreover, our regional-centred experience may limit the extrapolation of the results. Nevertheless, the evaluation of training sessions is very well documented and our JSPE results are consistent with other studies in French cohorts [12,13]. In our cohort we also confirmed differences in JSPE according to gender reported by other studies [9,14]. We had chosen not to use standardised patients. What can be seen as a limitation, appeared to us as a strength. Indeed, trainees reported that having to play this role helped them put themselves in the patient's shoes.

The training session helped participants take a step back from their clinical practice and reflect on kidney failure diagnosis delivery. Mounting evidence point the relationship between physicians' empathy as indicators of clinical competence and positive patient outcomes [9,11,15]. A recent study has shown that empathic comments significantly decrease the intensity of pain and modulate the activation of brain regions and key nodes of the pain system [16]. In addition, empathetic care has beneficial effects on doctors' mental health, as burnouts are reduced with higher empathy scores [13,17]. Hence, by improving trainees empathy and raising their awareness of the essential healing capacity and importance of listening skills, this training session might have a major impact and improve significantly the global quality of care.

Our study is one of the rare studies to assess teaching of the physician-patient relationship in nephrology. The strength of our cross-sectional and case control study is that it evaluated both a change of behaviour, assessed by a quantitative scale (i.e. JSPE) and a self-reported change of clinical practice in participants. Of note, studies dealing with levels 3 (change of behaviour) and 4a (self-reported change in professional practice) of the Kirkpatrick classification [8,18] are uncommon in the nephrology community worldwide [8]. Unlike in France and other European countries, communication skills training have become a relatively standard part of the curriculum in many United States medical schools. Likewise many nephrology fellowships in the US provide formal and informal training in communication skills, including breaking of the bad news. Along with this, the little available literature on the subject comes from the US and our study is, to our knowledge, the first European study on the subject. The study from Cohen *et al* on communication skill courses with first year nephrology trainees has reported promising results [19]. Indeed, the 26 participants of this daylong course reported improvement in communication skills and attitudes related to discussing disease progression, dialysis therapy withdrawal and end of life. NephroTalk, a three days communication skills curriculum for US nephrology fellows, appeared to increase their communication skills significantly in delivering bad news leading to more efficient encounters. Along with our study, communication skills' teaching seems essential in the nephrology curriculum [20,21]. Another new and interesting point of our study is the evaluation of empathy scores using the same JSPE scores in senior nephrologists. Scores appeared variable and only female gender was associated with significantly higher scores. Along with the absence of significant difference between

untrained/pre-training trainees and senior nephrologists, our study supports the initiation of a training session as a continuous medical education course. The notion that there is room for improvement in communication skills and empathy in senior nephrologists is also fairly new and challenges the established beliefs about how well senior practitioners do handle patient communication.

Role-play and simulation appeared to us the best and most effective way for the training of kidney failure diagnosis delivery. Simulation is used for the purpose of health professions education, either to teach technical gesture [22–25] or theoretical medical knowledge [26–29]. Simulation has also successfully been used to improve interprofessional socialisation [30,31]. Simulation is particularly useful in regards to skills coming into play when announcing bad news to the patient and his/her family [32–35]. Because setting up this training is both easy and low cost, we believe it can be replicated everywhere in the world, with a most likely high impact on participants. However educators have to keep in mind that simulation requires risk taking on the part of the learner. Therefore small group teaching and development of a safe learning environment is critically important [35,36].

Diagnosis of kidney failure and initiation of dialysis is a traumatic experience for patients. As analysed by G. Hercz, the evolution to dialysis therapy is a turbulent transition that may result in fragmented behaviour and thoughts [6]. Hence, we need to begin shifting the culture of dialysis care delivery, from a disease-centred approach, focused on clinical and laboratory disturbances, to a patient-centred approach, acknowledging the patients' need for support. Moreover, it is well documented that the physician-patient relationship and empathy of the physician impacts patients' care [4]. Physicians with greater empathy are more likely to achieve a better physician-patient relationship, with higher patient satisfaction, adherence to treatments, and health outcomes. In the context of medical education, it is therefore essential to understand and acknowledge the role of teaching in enhancing empathy in medical students. Role-playing appears to be an excellent way of improving perspective-taking skills and one's ability to put themselves in patients' shoes.

## Conclusion

It is possible to learn empathy and listening skills. These should therefore be taught in the medical curriculum, since they have a significant impact on patients' care. Role-play and simulation are an easy and effective way of teaching it. It helps the participant take a step back from their day-to-day practice.

## Supporting information

**S1 File. Evaluation AD final anonimized 0528.**
(XLSX)

**S2 File. Questions for residents final.**
(XLSX)

**S3 File. Questions for seniors final anonimized 0511.**
(XLSX)

**S4 File.** S1 Item. Satisfaction and impact survey at the end of the session. S2 Item. Long-term impact survey (spring 2019).
(DOCX)

## Acknowledgments

The authors would like to thank France Rein Ile de France Association for the participation in the project. They thank Naouel Zenaidi and Julie Plichon for her assistance in proofreading and editing this article.

## Author Contributions

**Conceptualization:** Alice Doreille, Eve Vilaine, Xavier Belenfant, Ziad Massy, Emmanuelle Corruble, Odile Basse, Dan Benhamou, Helene François.

**Formal analysis:** Alice Doreille, Helene François.

**Funding acquisition:** Xavier Belenfant, Emmanuelle Corruble, Odile Basse, Dan Benhamou, Helene François.

**Investigation:** Wided Tabbi, Yosu Luque, Eric Rondeau, Dan Benhamou, Helene François.

**Methodology:** Alice Doreille, Xavier Belenfant, Dan Benhamou, Helene François.

**Project administration:** Helene François.

**Supervision:** Helene François.

**Validation:** Alice Doreille, Eve Vilaine, Dan Benhamou, Helene François.

**Visualization:** Alice Doreille, Eve Vilaine, Wided Tabbi, Ziad Massy, Yosu Luque, Eric Rondeau, Dan Benhamou, Helene François.

**Writing – original draft:** Alice Doreille.

**Writing – review & editing:** Xavier Belenfant, Dan Benhamou, Helene François.

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
