## [Decision Letter · Decision Letter 0]

18 Mar 2021

PONE-D-20-29842

Can empathy be taught? A cross-sectional survey assessing training to deliver the diagnosis of end stage renal disease

PLOS ONE

Dear Dr. Francois,

Thank you for submitting your manuscript to PLOS ONE. After careful consideration, we feel that it has merit but does not fully meet PLOS ONE’s publication criteria as it currently stands. Therefore, we invite you to submit a revised version of the manuscript that addresses the points raised during the review process. Some minor comments need to be addressed for the article to be addressed.

We look forward to receiving your revised manuscript.

Kind regards,

Mohammed Saqr, Ph.D

Academic Editor

PLOS ONE

Journal Requirements:

2. Please include additional information regarding the survey or questionnaire used in the study and ensure that you have provided sufficient details that others could replicate the analyses.

For instance, if you developed a questionnaire as part of this study and it is not under a copyright more restrictive than CC-BY, please include a copy, in both the original language and English, as Supporting Information.

3. Please upload a copy of Supporting Information S1 item and S2 item, which you refer to in your text on page 14.

Reviewers' comments:

Reviewer's Responses to Questions

**Comments to the Author**

1. Is the manuscript technically sound, and do the data support the conclusions?

Reviewer #1: Yes

2. Has the statistical analysis been performed appropriately and rigorously? 

Reviewer #1: Yes

3. Have the authors made all data underlying the findings in their manuscript fully available?

Reviewer #1: Yes

4. Is the manuscript presented in an intelligible fashion and written in standard English?

Reviewer #1: Yes

5. Review Comments to the Author

Reviewer #1: This study evaluates an educational intervention for nephrology trainees in France to improve how the diagnosis of kidney failure is communicated with patients. This is a very novel study, and addresses an important problem in kidney disease care. It is well written and provides important information that can be used internationally to better train nephrologists.

I have one suggestion: Throughout the manuscript, it is recommended that the term “End-Stage Kidney Disease”- or better yet, kidney failure- be used instead of ESRD. See for example: Nomenclature for kidney function and disease: executive summary and glossary from a Kidney Disease: Improving Global Outcomes consensus conference - https://academic.oup.com/ndt/article/35/7/1077/5880573

6. PLOS authors have the option to publish the peer review history of their article (what does this mean?). If published, this will include your full peer review and any attached files.

Reviewer #1: No

---

## [Author Response · Author response to Decision Letter 0]

22 Mar 2021

Response to the reviewers

Please find enclosed a point by point answers to the reviewers' and editors' queries

Journal Requirements: 

We rechecked and confirm our manuscript meets PLOS ONE’s style requirements (title, main body, references and authors affiliations)

2. Please include additional information regarding the survey or questionnaire used in the study and ensure that you have provided sufficient details that others could replicate the analyses.

For instance, if you developed a questionnaire as part of this study and it is not under a copyright more restrictive than CC-BY, please include a copy, in both the original language and English, as Supporting Information.

The Jefferson Scale of empathy used in the study seems to be under copyright. Hence we unfortunately cannot provide a copy in Supporting Information.

“Copyright ©2001 by the authors. Health Policy Newsletter is a quarterly publication of Thomas Jefferson University, Jefferson Health System and the Office of Health Policy and Clinical Outcomes, 1015 Walnut Street, Suite 115, Philadelphia, PA 19107.”

(https://www.jefferson.edu/academics/colleges-schools-institutes/skmc/research/research-medical-education/jefferson-scale-of-empathy.html)

 3. Please upload a copy of Supporting Information S1 item and S2 item, which you refer to in your text on page 14.

Supporting information S1 item and S2 item were added to the manuscript. 

Abstract on the online submission form was amended 

We have reviewed our reference list and ensured it is complete and correct, with no duplication or retracted papers.

Reviewers' comments:

Reviewer's Responses to Questions

Reviewer #1: This study evaluates an educational intervention for nephrology trainees in France to improve how the diagnosis of kidney failure is communicated with patients. This is a very novel study, and addresses an important problem in kidney disease care. It is well written and provides important information that can be used internationally to better train nephrologists.

I have one suggestion: Throughout the manuscript, it is recommended that the term “End-Stage Kidney Disease”- or better yet, kidney failure- be used instead of ESRD. See for example: Nomenclature for kidney function and disease: executive summary and glossary from a Kidney Disease: Improving Global Outcomes consensus conference - https://academic.oup.com/ndt/article/35/7/1077/5880573

 Thanks a lot for this suggestion. We have replaced End Stage Renal Disease and ESRD by kidney failure throughout the manuscript.

---

## [Editor Report · Decision Letter 1]

29 Mar 2021

Can empathy be taught? A cross-sectional survey assessing training to deliver the diagnosis of end stage renal disease

PONE-D-20-29842R1

Dear Dr. Francois,

We’re pleased to inform you that your manuscript has been judged scientifically suitable for publication and will be formally accepted for publication once it meets all outstanding technical requirements.

Kind regards,

Mohammed Saqr, Ph.D

Academic Editor

PLOS ONE
---

## [Editor Report · Acceptance letter]

26 Aug 2021

PONE-D-20-29842R1 

Can empathy be taught? A cross-sectional survey assessing training to deliver the diagnosis of end stage renal disease 

Dear Dr. François:

I'm pleased to inform you that your manuscript has been deemed suitable for publication in PLOS ONE. Congratulations! Your manuscript is now with our production department. 

Kind regards, 

on behalf of

Dr. Mohammed Saqr 

Academic Editor

PLOS ONE